# Identification of Biomarkers and Molecular Pathways Implicated in Smoking and COVID-19 Associated Lung Cancer Using Bioinformatics and Machine Learning Approaches

**DOI:** 10.3390/ijerph21111392

**Published:** 2024-10-22

**Authors:** Md Ali Hossain, Mohammad Zahidur Rahman, Touhid Bhuiyan, Mohammad Ali Moni

**Affiliations:** 1Department of Computer Science and Engineering, Jahangirnagar University, Dhaka 1342, Bangladesh; ali.cse@diu.edu.bd (M.A.H.); rmzahid@juniv.edu (M.Z.R.); 2Health Informatics Lab, Department of Computer Science and Engineering, Daffodil International University, Dhaka 1216, Bangladesh; 3School of IT, Washington University of Science and Technology, Alexandria, VA 22314, USA; 4Faculty of Health and Behavioural Sciences, The University of Queensland, Brisbane 4072, Australia; 5Artificial Intelligence and Cyber Futures Institute, Charles Sturt University, Bathurst 2795, Australia

**Keywords:** lung cancer, COVID-19, smoking, comorbidity, protein-protein interaction, WGCNA, pathway analysis, ROC curve, survival analysis

## Abstract

Lung cancer (LC) is a significant global health issue, with smoking as the most common cause. Recent epidemiological studies have suggested that individuals who smoke are more susceptible to COVID-19. In this study, we aimed to investigate the influence of smoking and COVID-19 on LC using bioinformatics and machine learning approaches. We compared the differentially expressed genes (DEGs) between LC, smoking, and COVID-19 datasets and identified 26 down-regulated and 37 up-regulated genes shared between LC and smoking, and 7 down-regulated and 6 up-regulated genes shared between LC and COVID-19. Integration of these datasets resulted in the identification of ten hub genes (SLC22A18, CHAC1, ROBO4, TEK, NOTCH4, CD24, CD34, SOX2, PITX2, and GMDS) from protein-protein interaction network analysis. The WGCNA R package was used to construct correlation network analyses for these shared genes, aiming to investigate the relationships among them. Furthermore, we also examined the correlation of these genes with patient outcomes through survival curve analyses. The gene ontology and pathway analyses were performed to find out the potential therapeutic targets for LC in smoking and COVID-19 patients. Moreover, machine learning algorithms were applied to the TCGA RNAseq data of LC to assess the performance of these common genes and ten hub genes, demonstrating high performances. The identified hub genes and molecular pathways can be utilized for the development of potential therapeutic targets for smoking and COVID-19-associated LC.

## 1. Introduction

Lung cancer (LC) is a major public health problem around the world that has mainly been associated with smoking. Furthermore, recent epidemiological evidence has raised alarms about the heightened vulnerability of smokers to COVID-19, introducing a novel layer of concern. Thus, understanding the intricate interplay of smoking, COVID-19, and lung cancer is pivotal for devising comprehensive preventive and therapeutic strategies. For this context, this study employs advanced bioinformatics and machine-learning techniques to delve into the intricate relationship between these factors. Smoking’s pronounced association with COVID-19 [1,2,3,4,5,6,7], as elucidated in our recent work [3], sets the stage for this investigation.

Moreover, LC patients are more susceptible to COVID-19 and are more likely to suffer its more severe forms [8,9]. In fact, some recent studies found that COVID-19 was severe in patients with LC [8,9,10]. Notably, among them, one study identified hub genes that are associated with both COVID-19 and lung cancer [9]. In another study, Tania et al. [11] utilized a machine learning approach to identify key factors associated with COVID-19 infection, including smoking habits. Similarly, Sharifi et al. [12] demonstrated that machine learning models can accurately predict mortality risk in COVID-19 patients with a history of smoking. Moreover, Ali et al. [3] identified 9 hub proteins shared between COVID-19 and smoking, employing a machine learning approach to reveal key connections. Additionally, another study by Ma et al. [13] identified genes to classify smoking status in lung cells using machine learning. Smoking is a risk factor for severe COVID-19 [1,2,3,4,5,6,7] and lung cancer [14,15]. Thus, smoking not only emerges as a critical risk factor for severe COVID-19 but also establishes intricate links between COVID-19, and lung cancer. Furthermore, studies exist that explore the relationships between COVID-19 and lung cancer [9,10], smoking and COVID-19 [1,2,3,4,5,6,7], smoking and lung cancer [14,15].

However, there are no studies on smoking, COVID-19 and LC. Furthermore, several studies have already underscored the necessity of exploring the intricate relationships among lung cancer, smoking, and COVID-19 [5,8,16]. Against this backdrop, our study aims to unravel the complex interplay connecting these three elements, shedding light on their interdependencies and potentially providing crucial insights for better management and mitigation of their impact. To achieve this, an investigation at the molecular level using bioinformatics and machine learning approaches is essential to understand these interactions. Through robust statistical methodologies, we aimed to identify shared significant genes of COVID-19 and smoking with lung cancer, shedding light on potential molecular links among these factors and their impact on lung cancer. In this research, the Weighted Gene Co-expression Network Analysis (WGCNA) methodology, introduced by Langfelder and Horvath [17], was employed to identify correlated genes within clusters. WGCNA is well-established for assessing strengthened gene relationships [18]. The study adopts a network biology approach [19,20] to uncover commonly deregulated pathways and molecular signatures shared among smokers, COVID-19 and lung cancer.

By utilizing gene expression data from COVID-19-infected lung epithelial cells, a separate dataset from smoking lung epithelial cells, and datasets from lung cancer, the research identifies common Differentially Expressed Genes (DEGs) and overlapping pathways. Furthermore, the study employs WGCNA to construct correlation networks, shedding light on the relationships among these common genes.

In this study, RNAseq data from COVID-19 and smoking (SMK) were analyzed using the edgeR package. In contrast, microarray data from lung cancer (LC) were assessed using the limma R package to pinpoint disease-specific DEGs. Subsequently, significant DEGs common to all three diseases were identified. Furthermore, examining the correlation of these genes with patient outcomes through survival curve analyses [21] and performing gene ontology and pathway analyses will enhance our comprehension of the underlying biology. Additionally, utilizing mRNAseq data from The Cancer Genome Atlas (TCGA), lung cancer data were obtained and processed to identify genes common to all three conditions, as well as hub genes. Classification algorithms were employed to assess the identified genes. Finally, a WGCNA was executed on the significant genes to explore correlations among them. This comprehensive analysis aimed to unravel the intricate connections between COVID-19, smoking, and lung cancer, thereby enhancing our understanding of their interplay and potential implications. This insight may then guide the development of targeted interventions, facilitating precision medicine strategies.

## 2. Materials and Method

In this research, we used RNAseq data of COVID-19 and smoking as well as microarray data of LC from the publicly available NCBI Gene Expression Omnibus (GEO) (http://www.ncbi.nlm.nih.gov/geo/, accessed on 15 June 2024). We also utilized mRNAseq data of Lung Adenocarcinoma (LC) from publicly available The Cancer Genome Atlas (TCGA) through the TCGA genome data analysis centre (http://gdac.broadinstitute.org/, accessed on 15 June 2024). Figure 1 represents the workflow of our work, providing a clear overview of the steps and methodologies employed throughout the study.

We completed our research work using the following steps:DEG Identification: Utilizing the edgeR package for COVID-19 and SMK RNAseq data, and the limma R package for LC microarray data, differentially expressed genes (DEGs) were identified for each disease.Common DEG Identification: The common significant DEGs across the three diseases were identified by overlapping the DEGs obtained from COVID-19, SMK, and LC datasets.Pathway and GO Analysis: Pathway and Gene Ontology (GO) analyses were conducted on the identified common significant DEGs to unveil shared biological pathways and functional categories.PPI Network Construction: A Protein-Protein Interaction (PPI) network was constructed around the common genes to identify highly connected hub proteins using Cytoscape and Cyto-Hubba plugin.Survival Analysis of Hub genes: We also performed the survival analysis of hub genes to check their effect on the patient’s survival.TCGA Data Utilization: mRNAseq data for LC was obtained from the TCGA genome data analysis center. Cases (n = 510) with mRNA gene expression data were extracted, covering 20,510 genes.Data Preprocessing: Using the TCGA barcode, normal, tumor, and control samples were categorized based on barcode positions. Missing value samples and lower read-count (counts < 100) samples were removed. Data was normalized using the FPKM method.Common Gene FPKM Identification: FPKM values of common genes (COVID-19, SMK, and LC) and hub genes were extracted from the LC dataset.Performance Evaluation: The extracted common and hub gene FPKM values were subjected to classification algorithms to evaluate their performance.WGCNA Analysis: WGCNA was performed on significant genes to explore correlations among them using the WGCNA package.

### 2.1. Materials

We utilized RNAseq datasets of COVID-19 and SMK with the accession numbers GSE147507 and GSE47718, respectively, as well as microarray datasets of LC with the accession numbers GSE89039 and GSE136043 [22]. These datasets were downloaded from the NCBI Gene Expression Omnibus (GEO) database (http://www.ncbi.nlm.nih.gov/geo/, accessed on 15 June 2024) [23,24].

The GSE147507 dataset contains RNA-seq transcriptomics data obtained from lung cells infected with SARS-CoV-2 and controls (mock) within a 24-h time frame. This dataset was used to explore the gene expression changes induced by the virus. The GSE47718 dataset is RNA-seq data from the airway epithelium of healthy nonsmokers and smokers, providing insights into the impact of smoking on gene expression in the airway. We also utilized mRNAseq data of LC from TCGA through the TCGA genome data analysis centre (http://gdac.broadinstitute.org/, accessed on 15 June 2024). The specific dataset used was Lung Adenocarcinoma, TCGA PanCancer Atlas, containing mRNA gene expression information from 510 LC cases, with data on 20,510 genes.

### 2.2. Methods

To identify DEGs in COVID-19 and SMK datasets, we employed the edgeR package, applying a significance threshold of |logFC|≥1 and *p*-value < 0.05. Similarly, we utilized the Limma package for standard statistical procedures for the LC datasets, encompassing filtering, normalization, and performing Student’s unpaired t-test to identify DEGs, using the same significance threshold of |logFC|≥1 and *p*-value < 0.05. Through these analyses, we aimed to highlight genes with significant expression changes in relation to the respective conditions.

By employing these robust statistical methodologies on these datasets, we aimed to identify shared significant genes of COVID-19 and SMK with LC, shedding light on potential molecular links among these factors and their impact on lung cancer.

### 2.3. Proteomic Signatures: Construction and Analysis of Protein-Protein Interaction to Identify Hub Proteins

To delve deeper into the molecular interactions and potential key players within the identified common genes, we constructed a PPI network. This network provides valuable insights into the functional relationships and collaborations among proteins associated with LC, SMK, and COVID-19.

For this purpose, we utilized the STRING database (https://string-db.org/, accessed on 15 June 2024) to gather critical PPI information regarding the overlapping DEGs associated with LC and SMK and COVID-19 [19,25]. The STRING database integrates diverse interaction sources, including data derived from PubMed abstracts, co-expression patterns, gene fusion events, and neighbourhood-based associations. To ensure the reliability of interactions, a combined score exceeding medium confidence (>0.4) was selected as the threshold.

To visually represent the intricate PPI network, we employed Cytoscape (v3.9.1) with the Cyto-Hubba plugin [19,26], which offers algorithms such as degree, edge percolated component (EPC), maximal clique centrality (MCC), and maximum neighbourhood component (MNC) to identify highly interconnected proteins within the network [27]. For our analysis, we used MCC algorithm. This visualization not only facilitates the understanding of the interplay between proteins but also helps pinpoint hub proteins, which hold significance in driving the biological mechanisms underlying the complex relationship between LC, and associated factors like SMK and COVID-19.

### 2.4. Survival Analysis of Hub Genes

Subsequently, survival curves for significant hub genes were generated using an online KM plotter software (https://kmplot.com/analysis/, accessed on 15 June 2024) [28] to assess their potential influence on patient survival with the studied diseases or conditions.

### 2.5. Pathway and GO Analysis

Pathway and Gene Ontology (GO) analyses were conducted on the identified common significant DEGs to unveil shared biological pathways and functional categories. We used ClusterProfiler R package for these analyses [29].

### 2.6. Performance Evaluation of the Significant Genes with Classification Algorithms and WGCNA Analysis

By analyzing the TCGA barcode, we distinguished between normal and tumour samples based on the two digits at positions 14–15. Normal samples were denoted by digits 10 to 19, tumour samples by digits 01 to 09, and control samples by digits 20 to 29. We then removed samples with missing values and those with low read counts (total read count < 100). Subsequently, we employed the FPKM (Fragments Per Kilobase of transcript per Million mapped reads) method to normalize the dataset.

In this study, we focused on common genes of COVID-19 and smoking with LC. By extracting FPKM values of these common genes and hub genes from the dataset of 20,510 genes, we facilitated the evaluation of their performance using classification algorithms, including Bayesian Network, support vector machine (SVM), random forest (RF), and Logistic Regression. Additionally, we conducted WGCNA analysis on the significant genes to explore correlations among them, using the WGCNA package. This comprehensive approach aimed to uncover potential associations and interactions among the shared significant genes across COVID-19, smoking (SMK), and LC.

## 3. Result

### 3.1. Identification of Differentially Expressed Genes in COVID-19, Smoking and Lung Cancer

In the analysis of gene expression profiles, we identified a total of 1652 differentially expressed genes (DEGs) in the GSE136043 dataset associated with LC, out of which 765 were down-regulated and 887 were up-regulated. Similarly, in the GSE89039 dataset, we found 4028 DEGs, with 1369 genes being down-regulated and 2659 up-regulated in LC. In the context of COVID-19, we detected 739 DEGs, consisting of 353 up-regulated and 386 down-regulated genes. Furthermore, in smoking-associated data (SMK), 3866 DEGs were identified, with 1916 up-regulated and 1950 down-regulated genes.

Interestingly, we identified 26 down-regulated and 37 up-regulated genes shared between LC and SMK. Additionally, we observed 7 down-regulated and 6 up-regulated genes shared between LC and COVID-19 datasets. This cross-analysis of gene expression patterns provides valuable insights into the potential shared molecular mechanisms underlying the connections between LC, SMK, and COVID-19. Such shared genes could be critical players in the complex interplay between these factors and might hold the key to understanding their combined effects on disease outcomes.

### 3.2. Proteomic Signatures: Construction and Analysis of Protein-Protein Interaction to Identify Hub Proteins

In this study, we successfully identified a set of 76 significant common DEGs shared among SMK, COVID-19, and LC. Leveraging the STRING database, we investigated the protein-protein interactions (PPIs) among these 76 DEGs, leading to the construction of a comprehensive PPI network as depicted in the Figure 2. Utilizing Cytoscape and the Cyto-Hubba plugin, we conducted an in-depth analysis of the PPI network to uncover hub proteins with significant connectivity and importance within the network. Remarkably, our investigation highlighted the presence of ten hub genes, namely SLC22A18, CHAC1, ROBO4, TEK, NOTCH4, CD24, CD34, SOX2, PITX2, and GMDS. These hub genes emerge as crucial players in the intricate interplay of SMK, COVID-19, and LC. The identification of these hub genes suggests potential key regulatory points and intricate molecular interactions that could drive the relationship between smoking, COVID-19 susceptibility, and the development of LC. This study contributes to a deeper understanding of the molecular underpinnings connecting these complex health factors, offering insights that could pave the way for novel therapeutic strategies and improved clinical interventions.

### 3.3. Survival Analysis of Hub Genes

Survival analysis of lung cancer was conducted using an online KM plotter software. The input for this analysis comprised 295 patients with a smoking history. Array quality control involved excluding outlier arrays (n=2628), followed by multivariate analysis, which identified 4 genes (TEK, GMDS, CHAC1, CD24) as significant among the 10 hub genes. Additionally, for 1411 patients in the Univariate analysis, array quality control excluded biased arrays (n=2850), leading to the identification of 7 genes (CHAC1, TEK, CD24, SOX2, ROBO4, IMPT1 / SLC22A18, and GMDS) (see Figure 3) as significant among the 10 hub genes.

### 3.4. Pathway and GO Analysis

Pathway and Gene Ontology (GO) analyses were conducted on the identified common significant DEGs to unveil shared biological pathways and functional categories. We used ClusterProfiler R package for these analyses [29]. We identified the top 10 Kegg pathways (see Figure 4A) for the shared genes and significant enrichment terms (see Figure 4B) for these shared genes.

### 3.5. Performance Evaluation of the Significant Genes with Classification Algorithms and WGCNA Analysis

We explored the clustering patterns of the identified common genes using the WGCNA package in R. This analysis allowed us to uncover the inter-connectedness and co-expression patterns among these genes, revealing strong correlations among the hub genes (See Figure 5). Furthermore, we employed machine learning algorithms on the FPKM values of LC to evaluate the predictive performance of both the shared genes and the hub genes (see the ROC curve in Figure 6). The results of these analyses illustrated high-performance levels (See Table 1), indicating that these genes could serve as significant indicators or predictors in the context of LC and its related factors. This integrated approach involving WGCNA and machine learning algorithms contributes to a comprehensive understanding of the molecular associations and predictive power of the identified genes in the context of LC, smoking, and COVID-19.

## 4. Discussion

The convergence of COVID-19, smoking, and lung cancer is particularly significant due to the shared pathophysiological links, potentially leading to aggravated disease outcomes. The overlap of DEGs among these conditions hints at a complex interplay of cellular processes disrupted by both viral infection and smoking-induced damage, further fueling lung cancer susceptibility. It also underscores the importance of a holistic understanding of disease interactions for comprehensive patient care.

The integration of WGCNA and machine learning approaches strengthens the credibility of our findings. The robustness of the hub gene identification across different analytical methodologies reinforces their potential relevance. Additionally, the pathway analysis provides mechanistic insights into how these genes might modulate disease progression, offering directions for targeted therapeutic interventions. Several notable pathways across shared genes, including Hematopoietic cell lineage, Fructose and mannose metabolism, Biosynthesis of nucleotide sugars, Amino sugar and nucleotide sugar metabolism, Glutathione metabolism, Notch signaling pathway, Rheumatoid arthritis, and TGF-beta signaling pathway.

The identification of hub genes (SLC22A18, CHAC1, ROBO4, TEK, NOTCH4, CD24, CD34, SOX2, PITX2, and GMDS) and shared genes that bridge the relationship between COVID-19, smoking, and lung cancer holds profound implications for both understanding disease mechanisms and designing effective therapeutic strategies.

These identified hub genes may serve as attractive targets for drug development or indicators for disease prognosis. For instance, CHAC1 has been found to be upregulated in lung cancer [30]. Additionally, a study by Zhou et al. [31] demonstrated that the promotion of LC progression occurs through the degradation of glutathione (GSH) by MIA3 binding to CHAC1.

In a study investigating gene expression patterns in non-smoking lung cancer patients, researchers identified ROBO4 and TEK as down-regulated genes compared to healthy individuals [32], suggesting a potential role for these genes in the development or progression of lung cancer in non-smokers. Interestingly, contrary to expectations, increased expression of ROBO4 has been associated with improved overall survival in patients with non-small-cell lung cancer (NSCLC) [33], indicating a potential protective role for ROBO4 in NSCLC progression. Under physiological conditions, the TEK and ROBO4 receptor/ligand systems play critical roles in promoting vascular stability [34]. However, clinical studies have demonstrated downregulation of ANGPT1 and TEK expression levels in non-small-cell lung cancer (NSCLC) tissue [34,35]. Furthermore, the hub gene NOTCH4 has been implicated in both lung cancer and COVID-19. In COVID-19 patients, increased Notch4 expression on regulatory T cells is associated with disease severity and mortality, and its inhibition can rescue disease morbidity and mortality [36]. NOTCH signaling, including NOTCH4, may play a central role in SARS-CoV-2 viral entry, the inflammatory response, and lung regeneration [37]. In lung adenocarcinoma, high expression of NOTCH4 and C-X-C chemokine receptor 4 is predictive of poor prognosis [38]. The role of NOTCH4 in non-small cell lung cancer progression remains to be fully defined [39]. The study by Zarn et al. (1996) [40] found a significant association between the CD24 gene and lung cancer, showing that CD24 is physically linked with the c-fgr and lyn kinases in small cell lung cancer, suggesting a role in signaling. Expanding on this, Qiao (2021) [41] demonstrated that high expression of CD24, along with Hsp70 (a molecular chaperone), is associated with poor survival in lung cancer. Additionally, the expression of CD34, a marker for hematopoietic stem and progenitor cells, has been found in small cell lung cancer [42]. Karimi-Busheri (2013) [43] further identified the CD24+/CD38− phenotype as a potential biomarker for non-small cell lung cancer, collectively suggesting that CD24 plays a significant role in lung cancer development and progression.

Furthermore, the hub genes SOX2 and PITX2 have been associated with lung cancer and smoking. SOX2 overexpression has been linked to favourable outcomes in lung squamous cell carcinomas (Zheng, 2015) [44], while nicotine has been shown to induce the expression of SOX2 in non-small cell lung adenocarcinoma cells (Schaal, 2016) [45]. The up-regulation of PITX2 has been identified as an oncogenic mechanism in LC through the activation of the Wnt/β-catenin signalling pathway, as demonstrated in a study by Luo et al. (2019) [46]. This suggests that PITX2 may have potential as a novel diagnostic and prognostic biomarker in LC. Additionally, Nancy et al.’s research revealed that PITX2 exhibits low median expression levels across brain tissues in individuals with a smoking history [47]. Interestingly, PITX2 and ZFHX3, tied to cardioembolic stroke and atrial fibrillation, are linked to ICU admission in COVID-19 patients as observed in a study by Laia et al. [48].

One study suggests that increased GMDS expression is vital for cell proliferation and survival in LC, indicating its potential as a biomarker for diagnosis and treatment [49]. Moreover, the differential metabolic patterns of fructose and glucose in lung adenocarcinoma cells suggest a potential role for fructose in fueling the growth of these tumors [50].

While our study provides valuable insights, there are limitations. The study is based on bioinformatics analyses and requires experimental validation to confirm the functional roles of the identified genes. Moreover, individual variations and additional factors in disease progression should be considered for a comprehensive perspective. Additionally, another limitation of our study was the inability to conduct qRT-PCR analysis to validate the significance of the identified hub proteins, primarily due to the lack of laboratory resources.

In conclusion, our study bridges the gap between COVID-19, smoking, and lung cancer, shedding light on common molecular mechanisms. The hub genes discovered and their functional implications provide a foundation for further research and therapeutic innovations, potentially contributing to improved management and outcomes in patients with smoking and COVID-19-associated lung cancer.

## 5. Conclusions

In this study, we have successfully identified a set of hub genes (SLC22A18, CHAC1, ROBO4, TEK, NOTCH4, CD24, CD34, SOX2, PITX2, and GMDS) and common genes that establish a significant connection between COVID-19, smoking, and lung cancer. Survival analysis identified 7 genes (CHAC1, TEK, CD24, SOX2, ROBO4, IMPT1/SLC22A18, and GMDS) as significant and for the case of smoking patients, 4 genes (TEK, GMDS, CHAC1, CD24) were found as significant. These findings provide a profound understanding of the molecular interplay among these factors. By uncovering the roles and interactions of these genes, we have unveiled promising targets for therapeutic interventions and the development of personalized treatment strategies specifically tailored to lung cancer associated with smoking and COVID-19.

Furthermore, our analysis involved WGCNA and classification algorithms, yielding robust results that further validate the significance of these genes in the context of these diseases. Pathway analysis of the identified shared genes has illuminated the potential pathways through which these genes exert their influence, offering valuable insights into the underlying mechanisms.

The fusion of various omics data sources and the integration of advanced computational methods have enriched our comprehension of the intricate molecular landscape of these diseases.

Overall, our research acts as a groundwork for future investigations, stimulating the development of innovative therapeutic interventions and diagnostic approaches for improved outcomes in individuals with smoking and COVID-19-associated lung cancer.

### Key Points

We identified a set of hub genes (SLC22A18, CHAC1, ROBO4, TEK, NOTCH4, CD24, CD34, SOX2, PITX2, and GMDS) and common genes that establish a significant connection between COVID-19, smoking, and lung cancer.These genes can serve as biomarkers to detect early signs of lung cancer, especially in high-risk groups like smokers or individuals with COVID-19 history.Survival analysis revealed 7 significant genes (CHAC1, TEK, CD24, SOX2, ROBO4, IMPT1/SLC22A18, and GMDS), and for smoking patients, 4 genes (TEK, GMDS, CHAC1, CD24) were found significant.These findings provide a profound understanding of the molecular interplay among these factors and unveil promising targets for therapeutic interventions and the development of personalized treatment strategies tailored to lung cancer associated with smoking and COVID-19.By monitoring the expression of these genes, clinicians could stratify patients by risk (e.g., susceptibility to severe lung complications due to smoking or COVID-19) and tailor early interventions accordingly.Our analysis, including WGCNA and classification algorithms, yielded robust results that validate the significance of these genes in the context of these diseases.Pathway analysis of the identified shared genes illuminated the potential pathways through which these genes exert their influence, offering valuable insights into the underlying mechanisms.The fusion of various omics data sources and the integration of advanced computational methods enriched our comprehension of the intricate molecular landscape of these diseases.Our research serves as groundwork for future investigations, stimulating the development of innovative therapeutic interventions and diagnostic approaches for improved outcomes in individuals with smoking and COVID-19-associated lung cancer.

## Figures and Tables

**Figure 1 ijerph-21-01392-f001:**
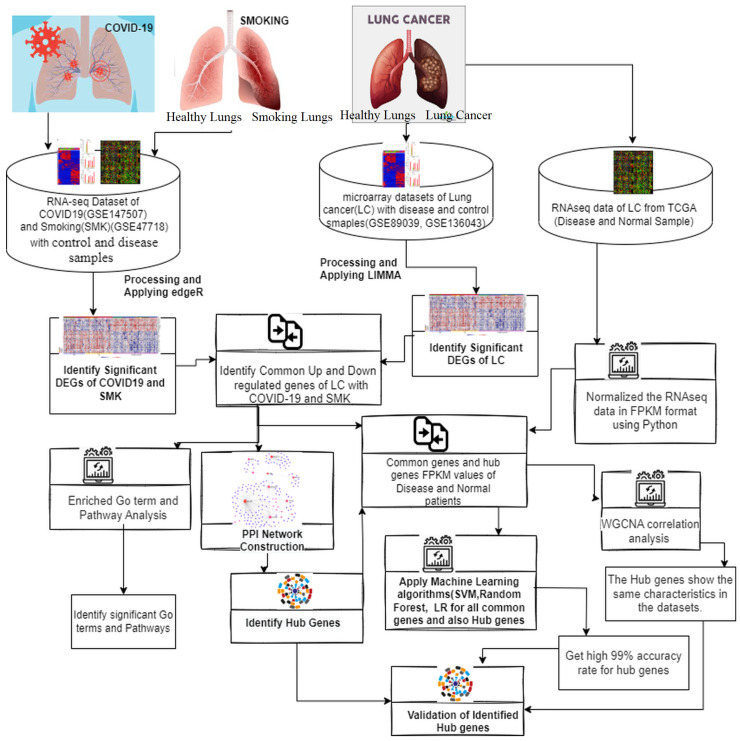
Workflow of the integrated analysis of the study. At first, RNAseq data of COVID-19 and SMK were processed using the edgeR package, while microarray data of LC were analyzed using the limma R package to identify disease-specific DEGs. Subsequently, common significant DEGs were identified across the diseases. Next, PPI network construction and analysis, Pathway and GO analysis were conducted on the shared DEGs to uncover hub proteins and shared pathways. After that, Utilizing mRNAseq data of LC from TCGA, mRNAseq data of LC was processed to identify mRNAseq data of shared genes and hub genes. The identified genes were then evaluated using classification algorithms. Additionally, a Weighted Gene Co-expression Network Analysis (WGCNA) was performed on significant genes to explore correlations between them. The entire analysis aimed to uncover connections between COVID-19, smoking, and LC, enhancing our understanding of their interplay.

**Figure 2 ijerph-21-01392-f002:**
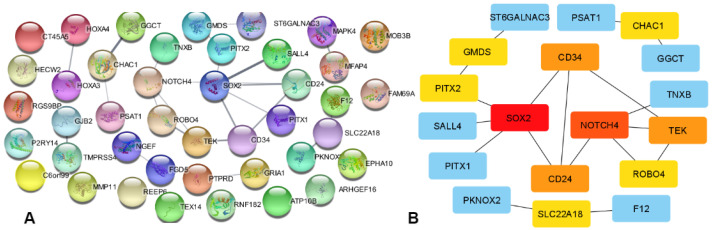
PPI Network for Significant genes of LC shared with smoking and COVID-19. (**A**) PPI network from string database. (**B**) 10 Hub genes identification through Cytoscape, Cyto-Hubba plugin and MCC algorithms.

**Figure 3 ijerph-21-01392-f003:**
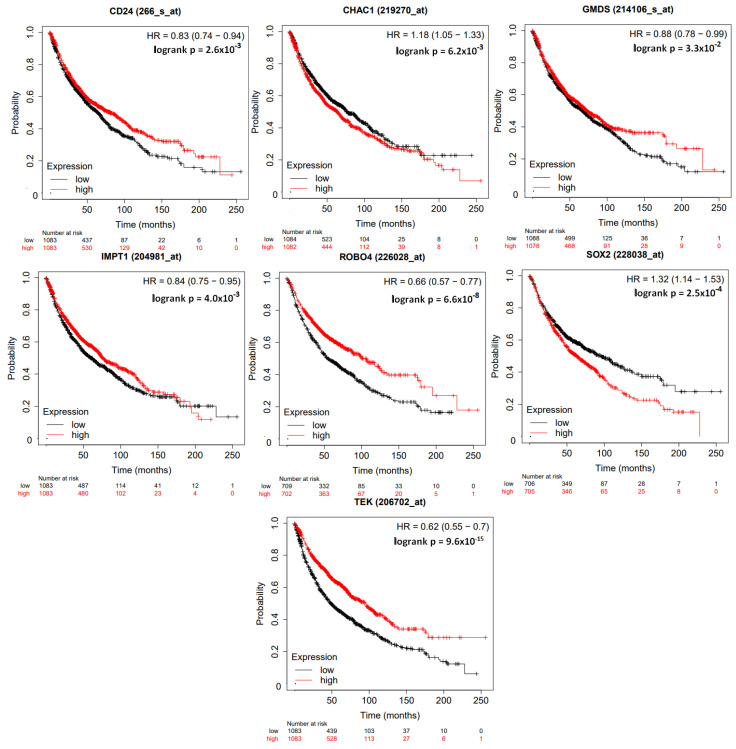
Survival curve of identified 7 significant genes (CHAC1, TEK, CD24, SOX2, ROBO4, IMPT1/SLC22A18, and GMDS) among 10 Hub genes.

**Figure 4 ijerph-21-01392-f004:**
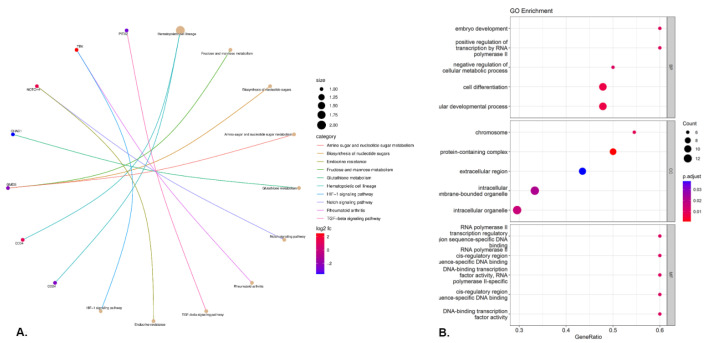
Functional enrichment analysis of shared genes. (**A**) Kegg Pathways (**B**) Gene Ontology.

**Figure 5 ijerph-21-01392-f005:**
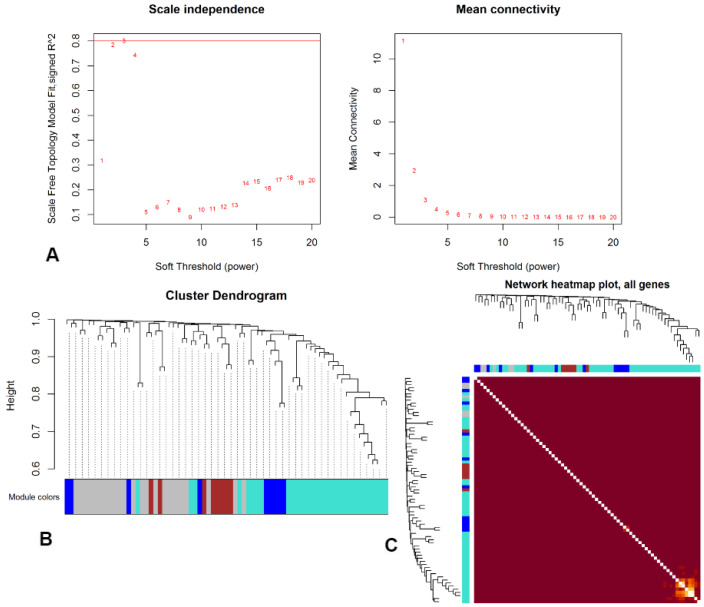
Correlation analysis using WGCNA for the significant genes of LC shared with COVID-19 and smoking. (**A**) Soft-threshold power selection plots for WGCNA analysis. The red numbers represent different tested power values (1–20), where power 5 was selected as the optimal threshold, achieving a scale-free topology model fit (R^2^ ≥ 0.8) while retaining a reasonable mean connectivity level. (**B**) Cluster dendrogram showing hierarchical clustering of genes based on expression data, with the height representing the dissimilarity between gene clusters. The colored bar below represents different gene modules, where each color denotes a distinct group of co-expressed genes. (**C**) Network heatmap plot of all genes showing topological overlap, with red indicating stronger gene connectivity and dendrograms representing hierarchical clustering of genes.

**Figure 6 ijerph-21-01392-f006:**
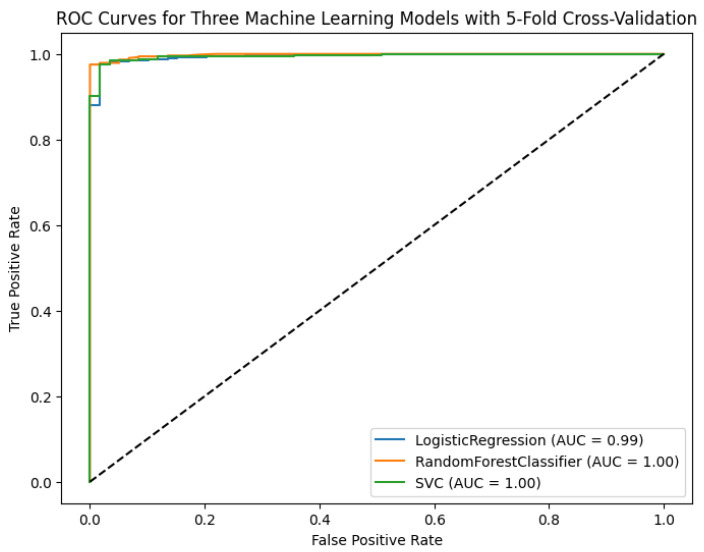
ROC curves were generated for hub genes to evaluate the predictive or diagnostic efficacy of these genes in identifying lung cancer.

**Table 1 ijerph-21-01392-t001:** Performance evaluation of the shared genes and hub genes.

PERFORMANCE EVALUATION OF SIGNIFICANT GENES				
**Name of the Algorithms**	**For 76 Shared Genes**		**For 10 Hub Genes**	
	**Train Accuracy**	**Test Accuracy**	**Train Accuracy**	**Test Accuracy**
Bayesian Network	1.00	1.00	0.990	0.988
Logistic Regression	1.000	0.991	0.982	0.979
Random Forest	1.000	0.997	1.000	0.983
SVM (Linear)	1.000	0.993	0.988	0.981

## Data Availability

The datasets used in our study were obtained from publicly available repositories, specifically the NCBI Gene Expression Omnibus (GEO) and The Cancer Genome Atlas (TCGA). Processed and analyzed data from our study will be made readily available for any inquiries. We utilized RNAseq datasets of COVID-19 and smoking with the accession numbers GSE147507 https://www.ncbi.nlm.nih.gov/geo/query/acc.cgi?acc=GSM4486165 (accessed on 15 June 2024) and GSE47718 https://www.ncbi.nlm.nih.gov/geo/query/acc.cgi?acc=GSE47718 (accessed on 15 June 2024), respectively, as well as microarray datasets of LC with the accession numbers GSE89039 https://www.ncbi.nlm.nih.gov/geo/query/acc.cgi?acc=GSE89039 (accessed on 15 June 2024) and GSE136043 https://www.ncbi.nlm.nih.gov/geo/query/acc.cgi?acc=GSE136043 (accessed on 15 June 2024). We utilized mRNAseq data of Lung Adenocarcinoma (LC) https://www.cbioportal.org/study/summary?id=luad_tcga_pan_can_atlas_2018 (accessed on 15 June 2024) from TCGA through the TCGA genome data analysis centre (http://gdac.broadinstitute.org/ (accessed on 15 June 2024)).

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
