# Peer review of "Identification of Biomarkers and Molecular Pathways Implicated in Smoking and COVID-19 Associated Lung Cancer Using Bioinformatics and Machine Learning Approaches"

_ijerph, 2024, doi:10.3390/ijerph21111392_

Round 1
Reviewer 1 Report
Comments and Suggestions for Authors
Identification of Biomarkers and Molecular Pathways Implicated in Smoking and COVID-19 Associated Lung Cancer Using Bioinformatics and Machine Learning Approaches.
General comments
The study employed advanced bioinformatics and machine-learning techniques to delve into the intricate relationship between the genes in covid-19, lung cancer(LC) and smoking. The authors compared the differentially expressed genes (DEGs) between LC, smoking, and COVID-19 datasets and identified 26 down-regulated and 37 up-regulated genes shared between LC and smoking, and 7 down-regulated and 6 up-regulated 21 genes shared between LC and COVID-19.
Specific comments
Abstract
Page 2, line 50: cancer [ 9, 10], smoking and COVID-19, Smoking and Lung cancer – using lower case letters for smoking and lung cancer.
2. Materials and Methods
This section requires an opening statement indicating how the data sets were obtained, whether specific approvals were required and whether there were ethical issues related to the use of the data.
There should be a subsection named ‘Materials’ before the one called ‘Methods’
Figure 1 should come under the subsection ‘Methods’
Page 3, line 87: ‘Our research work is completed using the following steps:’ Use past tense consistently in this section
Comments on the Quality of English LanguageThe English requires only minor editing-the teat in materials and methods should be in past tense and capital letters should be used appropriately.
Author Response
1. Summary
Thank you very much for taking the time to review this manuscript. Please find the detailed responses below and the corresponding revisions/corrections highlighted in the re-submitted files.
3. Point-by-point response to Comments and Suggestions for Authors
Comments 1: [Page 2, line 50: cancer [ 9, 10], smoking and COVID-19, Smoking and Lung cancer – using lower case letters for smoking and lung cancer.]
Response 1: Thank you for pointing this out. We agree with this comment. Therefore, we have addressed your suggestions in our updated manuscripts [ page-2, paragraph:-1]
Comments 2: [Materials and Methods: This section requires an opening statement indicating how the data sets were obtained, whether specific approvals were required and whether there were ethical issues related to the use of the data]
Response 2: Thank you for your suggestions. We have updated our manuscripts according to your suggestions [page number: 4, paragraph:-1]. As we used the publicly available datasets of NCBI and TCGA repositories [page:17, paragraph:-7], there are no specific approvals or ethical issues.
Comments 3: There should be a subsection named ‘Materials’ before the one called ‘Methods’. Figure 1 should come under the subsection ‘Methods’.
Response 3: Thank you for your suggestions. We have added the subsection named ‘Materials’ before the one called ‘Methods’[page number:-5]. We have also shifted the Figure1 under the subsection ‘Methods’ [page number:-6].
Comments 4: Page 3, line 87: ‘Our research work is completed using the following steps:’ Use past tense consistently in this section.
Response 4: Thank you for your suggestion. We have updated the manuscripts according to your instruction [page:-4, paragraph:-2].
4. Response to Comments on the Quality of English Language
Point 1: The English requires only minor editing-the teat in materials and methods should be in past tense and capital letters should be used appropriately.
Response 1: Thank you for your nice suggestion. We have modified the manuscripts according to your suggestions.
Reviewer 2 Report
Comments and Suggestions for Authors
Providing more information on how the obtained findings can be integrated into clinical applications can enhance the practical value of the research. For instance, more details can be given on how the identified hub genes can be used as potential therapeutic targets.
Comments on the Quality of English LanguageI believe that the English used is at an acceptable level; however, I did not find the text to be fluent.
Author Response
- Summary
Thank you very much for taking the time to review this manuscript. Please find the detailed responses below and the corresponding revisions/corrections highlighted in the re-submitted files.
3. Point-by-point response to Comments and Suggestions for Authors
Comments 1: [Providing more information on how the obtained findings can be integrated into clinical applications can enhance the practical value of the research. For instance, more details can be given on how the identified hub genes can be used as potential therapeutic targets.]
Response 1: Thank you for your suggestions. We have added the possible clinical applications of these genes in the Key Points section at the end of the manuscripts (page: 16-17).
4. Response to Comments on the Quality of English Language
Point 1: I believe that the English used is at an acceptable level; however, I did not find the text to be fluent.
Response 1: Thank you for your nice comment and suggestion. We have modified the manuscripts according to your suggestions.
Reviewer 3 Report
Comments and Suggestions for Authors
Dear Editor,
the manuscript of Hossain M.A. et al. entitled Identification of Biomarkers and Molecular Pathways Implicated in Smoking and COVID-19 Associated Lung Cancer Using Bioinformatics and Machine Learning Approaches concerns a very current issue and provides relevant data about new genes which may become new candidates as biomarkers involved in the predisposition to lung cancer development when smoking habits and COVID-19 infection occur.
In this original article, the authors have presented the issue in a clear and well-written manner, structuring the manuscript appropriately.
As stated by the authors, this study can provide the basis for other studies, in order to confirm the actual involvement and role of these genes in this pathology. Indeed, this is a bioinformatic study and since my expertise is in the biomedical field; to increase the quality of the manuscript, I would advise the authors to consider one of the following options:
1) Involving a team of biologists and/or clinicians, include a small cohort of patients to verify your results. For example, you could include Next Generation Sequencing techniques, or other laboratory techniques, such as immunostaining or molecular investigations as western blots to be conducted on patient tissues.
2) In case the first option is not feasible, I would better discuss all genes, summarising a broader literature that supports or contradicts the present study (see lines 265 to 313).
Author Response
1. Summary
Thank you very much for taking the time to review this manuscript aw well as for the nice comments. Please find the detailed responses below and the corresponding revisions/corrections highlighted in the re-submitted files
Comments 1: [Involving a team of biologists and/or clinicians, include a small cohort of patients to verify your results. For example, you could include Next Generation Sequencing techniques, or other laboratory techniques, such as immunostaining or molecular investigations as western blots to be conducted on patient tissues.]
Response 1: These are the limitations of our work that are given in the discussion section [page:-15, Paragraph:-2]. Our current study focuses on computational analysis, and future work may involve experimental validation in collaboration with clinical experts to confirm the functional roles of the identified genes.
Comments 2: In case the first option is not feasible, It would better discuss all genes, summarising a broader literature that supports or contradicts the present study (see lines 265 to 313).
Response 2: We appreciate the reviewer's suggestion to discuss the identified genes in more depth. In response, we have expanded the discussion to include a broader review of literature that supports or contradicts our findings. For example, CHAC1 has been reported as upregulated in lung cancer, while ROBO4 and TEK are downregulated in non-smoking lung cancer patients, but ROBO4 is associated with improved survival in NSCLC. Furthermore, we explored the significance of NOTCH4 in both lung cancer and COVID-19, and discussed the roles of CD24, CD34, SOX2, PITX2, and GMDS in lung cancer progression. This expanded discussion now provides a more balanced interpretation of the hub genes identified in our study.
Round 2
Reviewer 2 Report
Comments and Suggestions for Authors
The references used in the study are generally appropriate and relevant, but the inclusion of some additional references could strengthen the paper. In particular, more recent studies that better support the biomolecular links between COVID-19, smoking, and lung cancer, or that explain the machine learning approaches in greater detail, could be beneficial.
Comments on the Quality of English Languagethe manuscript is well-written but could benefit from a language polishing.
Author Response
1. Summary
Thank you very much for taking the time to review this manuscript. Please find the detailed responses below and the corresponding revisions/corrections highlighted in the re-submitted files.
3. Point-by-point response to Comments and Suggestions for Authors
Comments 1:[The references used in the study are generally appropriate and relevant, but the inclusion of some additional references could strengthen the paper. In particular, more recent studies that better support the biomolecular links between COVID-19, smoking, and lung cancer, or that explain the machine learning approaches in greater detail, could be beneficial.]
Response 1: Thank you for your suggestions. In this context, we have incorporated recent studies on machine learning that support our work in uncovering shared genes and pathways among these factors, providing a more robust and up-to-date foundation for our investigation. [page: 2-3]
4. Response to Comments on the Quality of English Language
Point 1: The manuscript is well-written but could benefit from a language polishing.
Response 1: Thank you for your valuable suggestion. We have revised the manuscript with input from an Australian professor, Dr. Mohammad Ali Moni, and a U.S. professor, Dr. Touhid Bhuiyan. Additionally, we have made efforts to polish the language to enhance clarity and readability.